# Serosurvey in Two Dengue Hyperendemic Areas of Costa Rica Evidence Active Circulation of WNV and SLEV in Peri-Domestic and Domestic Animals and in Humans

**DOI:** 10.3390/pathogens12010007

**Published:** 2022-12-21

**Authors:** Marta Piche-Ovares, Mario Romero-Vega, Diana Vargas-González, Daniel Felipe Barrantes Murillo, Claudio Soto-Garita, Jennifer Francisco-Llamas, Alejandro Alfaro-Alarcón, Carlos Jiménez, Eugenia Corrales-Aguilar

**Affiliations:** 1Virology-CIET (Research Center for Tropical Diseases), Universidad de Costa Rica, San José 11501-2060, Costa Rica; 2PIET (Tropical Disease Research Program), Department of Virology, School of Veterinary Medicine, Universidad Nacional, Heredia 86-3000, Costa Rica; 3Department of Pathology, School of Veterinary Medicine, Universidad Nacional, Heredia 86-3000, Costa Rica; 4Laboratorio de Investigación en Vectores-CIET (Research Center for Tropical Disease), Universidad de Costa Rica, San José 11501-2060, Costa Rica; 5School of Nursing and Health Studies, University of Miami, Coral Gables, FL 248153, USA

**Keywords:** flavivirus, Costa Rica, WNV, SLEV, seroepidemiology

## Abstract

Costa Rica harbors several flaviviruses, including Dengue (DENV), Zika (ZIKV), West Nile virus (WNV), and Saint Louis encephalitis virus (SLEV). While DENV and ZIKV are hyperendemic, previous research indicates restricted circulation of SLEV and WNV in animals. SLEV and WNV seroprevalence and high transmission areas have not yet been measured. To determine the extents of putative WNV and SLEV circulation, we sampled peri-domestic and domestic animals, humans, and mosquitoes in rural households located in two DENV and ZIKV hyperendemic regions during the rainy and dry seasons of 2017–2018 and conducted plaque reduction neutralization test assay for serology (PRNT) and RT-PCR for virus detection. In Cuajiniquil, serological evidence of WNV and SLEV was found in equines, humans, chickens, and wild birds. Additionally, five seroconversion events were recorded for WNV (2 equines), SLEV (1 human), and DENV-1 (2 humans). In Talamanca, WNV was not found, but serological evidence of SLEV circulation was recorded in equines, humans, and wild birds. Even though no active viral infection was detected, the seroconversion events recorded here indicate recent circulation of SLEV and WNV in these two regions. This study thus provides clear-cut evidence for WNV and SLEV presence in these areas, and therefore, they should be considered in arboviruses differential diagnostics and future infection prevention campaigns.

## 1. Introduction

The genus *Flavivirus* is composed of single-stranded RNA viruses, which include Dengue virus (DENV), Saint Louis encephalitis virus (SLEV), Zika virus (ZIKV), West Nile virus (WNV), and Yellow Fever virus (YFV) [1,2]. All four of which are arboviruses that cause mosquito borne diseases throughout the Americas and are responsible for thousands of deaths and hospitalizations every year [3,4]. Many factors are recognized as contributing to their wide dissemination and re-emergence, e.g., poorly planned urbanization, geographical expansion of vectors, changing environmental conditions, and deforestation [5,6,7].

The transmission cycles of these viruses involve a wide variety of susceptible species such as humans, rodents, horses, birds, and nonhuman primates [8]. The clinical presentation of acute flavivirus infections in humans and vertebrates ranges from mild illness (e.g., asymptomatic infection) or self-limiting febrile episodes to severe and life-threatening diseases (hemorrhagic fever, shock syndrome, encephalitis, congenital defects) [9,10,11].

WNV and SLEV belong to the Japanese encephalitis serocomplex [12]. Both are neurotropic flaviviruses that can cause encephalitis, seizure disorders, and paralysis in humans and equines [10,13,14,15]. In the United States, several mosquito species have been shown to have high or moderate vectorial capacity, including *Culex tarsalis*, *Cx. pipiens*, *Cx restuans*, *Cx. quinquefasciatus*, *Cx. stigmatosoma*, and *Cx. nigripalpus* [16,17,18]. Additionally, migratory birds can serve as dispersal vehicles when they move seasonally and stop at different sites during their journey, establishing possible dispersal events [19]. Most mammals, such as equines and humans, are dead-end hosts because of the low-level viremia produced after infection [16,20].

Costa Rica is endemic for DENV [21,22]. Molecular epidemiological studies show broad circulation of DENV 1–3 in humans, and there is both molecular and serological evidence of DENV-4 circulation in wild animal samples [23,24]. Cocirculation of different flaviviruses adds complexity to clinical and laboratory diagnosis, because of significant cross-reactivity and similarities in the undifferentiated fever-like initial symptoms [25]. The National Health Service of Costa Rica does not currently include SLEV and WNV in their routine diagnostic panel for arbovirus detection; so, the epidemiology and/or local presence of these viruses in the human population is still poorly studied. In 2004–2005, serological evidence of contact with WNV was found in asymptomatic equines and sloths (*Choloepus hoffmanni* and *Bradypus variegatus*) in the regions of Guanacaste and Upala [26,27]. Additionally, sloths presented antibodies against SLEV [27]. In 2009, the first cases of clinical disease in equines associated with WNV were reported. After this first report, positive animals continue to be reported annually. Between 2009 and 2017, there were 32 cases of symptomatic equines positive for WNV [28]. Continuous monitoring in endemic areas, such as Costa Rica, and in other tropical areas for early detection and timely reporting are crucial to evaluate the risk of transmission to humans and animals [25,29]. In such areas, epidemiological surveillance based on regular sampling of equines, sentinel chickens, and wild birds has demonstrated good sensitivity [25,29].

Since the presence of zoonotic flaviviruses other than DENV and ZIKV has not been readily determined in humans in our country, the current study aimed to be a proof of concept of the cocirculation of WNV and SLEV in two rural areas that are hyperendemic for other human-infecting flaviviruses and aimed to detect putative reservoirs and vectors for these viruses. Humans, wild birds, equines, and mosquito samples were analyzed to better understand if the viral cycle was present in those areas. We sampled during the rainy and dry seasons of 2017–2018 and conducted plaque reduction neutralization tests (PRNT) for serology and reverse transcription polymerase chain reactions (RT-PCR) for virus detection. We reported, here, several seroconversion events in different peri-domestic and domestic species but found no evidence of active viral infection in any mosquito or bird samples. This seroconversion evidence supports recent circulation of SLEV and WNV in these two regions. This evidence must be taken into account for future prevention campaigns but, most importantly, in arboviruses differential diagnostics.

## 2. Materials and Methods

### 2.1. Study Area

The study was conducted in two regions of Costa Rica (i.e., Cuajiniquil and Talamanca) where previous flavivirus infections (DENV, ZIKV in humans, and WNV in horses) were officially recorded by Ministerio de Salud de Costa Rica (National Health Service) and the Servicio Nacional de Salud Animal (Animal Health Service), respectively [28,30,31]. Sampling was performed during the rainy and dry seasons of 2017 and 2018. Cuajiniquil is located on the Pacific coast (10°15′06″ N, 85°41′07″ O) in the province of Guanacaste in the northwestern part of Costa Rica (Figure 1A), while Talamanca (9°37′14.99″ N, 82°50′39.98″ O) is located on the South Caribbean coast (Figure 1B). At each study site, 8 households were chosen for sampling. The criteria for selecting households were: (i) the presence of at least one equine that was unvaccinated against WNV, (ii) a forest patch near the household (25–50 mts), and (iii) that the household inhabitants were willing to participate and signed an informed consent. Serum samples from equines, humans, chickens, and wild birds were taken. At the same time, wild birds were captured and identified using morphological examination. Sampling was performed twice at each household, i.e., once during the rainy (high arbovirus transmission rates) and once during the dry (low arbovirus transmission rates) seasons. The rationale for sampling each household twice at least 6 months apart was to detect putative flavivirus seroconversion events.

### 2.2. Sampling and Classification of Wild Birds

Birds were captured using mist-nets positioned at two sites (forest and peri-domiciliary) in each household. At least, five birds per household were taxonomically identified and then euthanized by an intramuscular anesthesia overdose (ketamine 10 mg/kg + xylazine 1 mg/kg) [32,33]. Blood samples were taken and stored at 4 °C until arriving at the laboratory and stored at −70 °C for later analysis. Additionally, samples of organs were aseptically collected (heart, lung, liver, spleen, intestine, kidney, brain, reproductive tract, eye, and proventriculus). A portion of these samples was conserved in RNA later (Thermo Scientific, cat AM0721) and another portion in 10% buffered formalin; additionally, a pool of organs was collected in RNA later for RT-PCR positivity initial screening. A complete postmortem and histopathological analysis were performed for each animal. Tissue samples were processed and embedded in paraffin based on standard procedures as described elsewhere [34]. Routine stained hematoxylin and eosin slides were analyzed to characterize any inflammatory infiltrates and for the identification/distribution of lesions.

### 2.3. Sampling and Classification of Mosquitoes

Field sampling of mosquitoes was conducted in parallel. Encephalitis vector survey (EVS) traps (BioQuip Products Inc., Compton, CA, USA) baited with CO_2_ were placed for 12–16 h in four different locations in each household: inside, peri-domiciliary, barn, and forest. Mosquitoes were collected the next morning and transferred to the field lab on ice. A taxonomical identification to species level was achieved using published keys [35,36]. Mosquitoes were pooled according to the collection site and species (maximum 20 individuals per pool) for RT-PCR virus detection. Blood-engorged females were analyzed individually to identify blood feeding sources.

### 2.4. Sampling of Equines, Humans, and Chickens

Blood samples from equines were taken by puncture of the jugular vein, only animals older than 6 months were sampled. Gender, age, breed, and travel history were recorded. Chicken (*Gallus gallus*) samples were taken from the wing vein. The human sample was taken from peripheral venipuncture after informed consent. Whole blood was centrifuged, and serum was stored at −20 °C for serological analysis.

### 2.5. Virus Strains

For PRNT analysis, different flavivirus-envelope-protein-expressing yellow fever chimeric viruses donated by the Center for Disease Control and Prevention were used, except for ZIKV, for which an American Type Culture Collection (ATCC) reference strain was utilized [37,38,39]. The strains used in this study were: WNV (YFV 17D/WNV Flamingo 383-99), DENV 1-4 (YFV 17D/DENV-1 PUO 359, YFV 17D/DENV-2 218, YFV 17D/DENV-3 PaH881/88, YFV 17D/DENV-4 1228), ZIKV (ATCC VR-748), SLEV (YFV 17D/SLEV CorAn 9124), and YFV (YFV 17D) [37,38,39].

### 2.6. Serological Screening by Plaque Reduction Neutralization Tests (PRNT)

Flavivirus exposure was evaluated in sera obtained from horses, humans, domestic chickens, and wild birds by PRNT, considered the gold standard for determining *Flavivirus* antibodies [40,41]. Serum samples were heat-inactivated at 56 °C for 30 min. Then, they were used for an initial screening against WNV and SLEV at a 1:10 dilution [41,42]. Briefly, samples were diluted 1:5 in minimum essential medium (MEM) with 2% of fetal bovine serum and mixed with an equal volume of each virus to an estimated result of 10 plaque formation units/well. The virus–antibodies mix was incubated 1 h at 37 °C in a 5% CO_2_ atmosphere; then, a 100 µL volume was inoculated into a Vero (ATCC CCL-81) cells monolayer previously seeded in a 48 well-plate and incubated for an hour. Then, it was removed and 500 µL of MEM with 2% of fetal bovine serum and 1% of carboxymethylcellulose were added. After 5 days of incubation, plates were fixed with formalin (3.7%) for an hour and stained with crystal violet (1%). Sera that resulted in 90% neutralization relative to the average of the viral control (no sera) were considered WNV or SLEV reactive. Due to smaller volumes of sera, wild bird and chicken samples were tested in a 96 well-plate format using a similar protocol and fixated after 3 days of incubation [42].

Samples from humans and equines that were considered reactive (90% reduction of foci) were tested in a serial two-fold dilution that ranged from 1:20 to 1:1280 against WNV-, DENV 1-4-, SLEV-envelope-protein-expressing yellow fever chimeric viruses, with ZIKV and YFV in similar conditions as the previously described protocol. Wild bird and chicken serum samples were only tested against WNV- and SLEV-envelope-protein-expressing yellow fever chimeric viruses because of the limited sera volume. A plaque reduction of ≥90% was considered positive, with the titer measurement as the highest serum dilution showing ≥90% of plaque relative to the average of the viral control. A 4-fold difference in titer among flaviviruses was required for unequivocally classifying a given serum sample as specifically neutralizing a particular flavivirus. In cases where there was less than 4-fold dilution difference, sera were classified only as “unspecific”.

### 2.7. Flavivirus RT-PCR in Wild Bird and Mosquito Samples

Viral RNA was extracted from avian tissue (pool of organs) and mosquito pools using the TRIzol (Ambion, Austin, TX, USA 15596018) method according to the manufacturer’s instructions. Reverse transcription was performed using a Revert Aid First Strand cDNA Synthesis Kit (Thermo Scientific, cat K1622) with random hexamers primers. A negative (water) and positive control (glyceraldehyde 3-phosphate dehydrogenase (GAPDH)) were included. Total RNA of the sample was quantified using a NanoDrop 2000 (Thermo Scientific, Waltham, MA, USA ND-2000).

First, a seminested PCR was performed using Flavivirus genus specific primers localized in the nonstructural protein 5 (NS5) following a previously described protocol [43]. A positive control (YFV 17D) and negative control (water) were included. PCR products were analyzed and quantified using QIAxcel DNA screening gel (Qiagen, Hilden, Germany 929554); a 220 base-pair band of cDNA was expected [43]. Positive samples from the nested PCR were purified using ExoSAP-IT (Applied Biosystems, Waltham, MA, USA 78201), following manufacturer instructions. Then, Sanger sequencing of both strands was performed by Macrogen Inc. (Seoul, South Korea). The resulting sequence was compared with entries in the GenBank database using the nucleotide basic alignment search tool (BLASTn) (https://blast.ncbi.nlm.nih.gov/Blast.cgi, accessed on 16 September 2020) and MEGA X software [44].

### 2.8. Mosquito Blood Meal Identification

To identify the mosquitoes’ blood meals, blood-engorged females were taxonomically identified and processed individually. Mosquitoes were macerated in a 1.5 mL tube and DNA-RNA was extracted using NucleoSpin TriPrep (740966.50, Macherey-Nagel). RNA that was obtained from these samples was analyzed for flaviviruses as previously described [43].

Blood meal identification was determined using a set of primers for cytochrome oxidase subunit I (COI), following the protocol of Townzen et al. 2008 [45]. PCR products were purified using ExoSAP-IT (Applied Biosystems, Waltham, MA USA 78201) and subjected to nucleotide sequencing with forward and reverse primers at Macrogen Inc. (Seoul, South Korea). The sequence was compared with entries in GenBank database using the nucleotide basic alignment search tool (BLASTn) (https://blast.ncbi.nlm.nih.gov/Blast.cgi, accessed on 16 September 2020) and MEGA X software [44].

### 2.9. Ethical Statement

The study, associated protocols, and sampling permits were written based on national ethical legislation and approved by the Institutional Committee of Care and Use of Animals of the University of Costa Rica (CICUA-042-17), Committee of Biodiversity of the University of Costa Rica (VI-2994-2017), National System of Conservation Areas (SINAC): Tempisque Conservation Area (Oficio-ACT-PIM-070-17), La Amistad-Caribe Conservation Area (M-PC-SINAC-PNI-ACLAC-047-2018). The survey did not involve endangered or protected species.

Informed consent was obtained from all subjects involved in the study. After signing of an informed consent previously approved by the University of Costa Rica´s Ethic Scientific Committee (CEC or IRB in English) (CEC-VI-4050-2017), a blood sample was taken from humans for serology analysis.

## 3. Results

### 3.1. Several Flaviviruses Cocirculate in Each Sampled Region

Serum samples were analyzed using serology in a PRNT ≥90% test. A total of 106 equines, 33 humans, 39 chickens, and 140 wild birds were tested at both sites. In order to record seroconversion events, the same equines (73/106) and humans (26/33) were sampled 6 months apart (Table 1 and Table 2).

At Cuajiniquil, Guanacaste, 37 (43%) of the equines, 1 bird (2%), 1 (3%) chicken, and 1 (6%) human had neutralizing antibodies against WNV (4-fold dilution of difference) (Figure 2). Additionally, serological evidence for SLEV was found in 11 (13%) equines, 1 (2%) wild bird, 1 (6%) human, and 1 (3%) chicken. This analysis also showed that 6 (38%) of the human samples had antibodies against DENV-1 (Figure 2B). A total of 5 of these positive samples were seroconversion events: 2 for WNV in horses (<20 in the first sampling for both and >1280, 640 in the second sampling), 1 for SLEV (<20 in the first sampling, 320 in the second sampling), and 2 for DENV-1 in humans (<20 in the first sampling for both and 320 and 680 in the second sampling). No serological evidence of DENV-2, -3, -4, ZIKV, and YFV was found. At this site, 18 horses (21%), 8 humans (50%), 30 chickens (94%), and 48 (92%) wild birds were negative for all the targeted viruses. In addition, 20 horses (23%) and 2 birds (4%) were positive for two or more flaviviruses and were classified as “unspecific”. The wild bird species with neutralizing antibodies against WNV was identified as *Campylorhynchus rufinucha*, a very common resident species in that area [32].

In contrast, we did not detect any serological evidence for WNV in Talamanca (Figure 2). However, evidence of previous contact with SLEV was found. Neutralizing antibodies against SLEV were found in 12 (60%) equines, 3 (18%) humans, and 2 (2%) wild birds (Figure 2). Additionally, 6 (35%) human samples were positive against DENV-1, and 1 (6%) had neutralizing antibodies against DENV-1 and DENV-2 (Figure 2B). At this site, 3 equines (15%), 7 humans (41%), and 84 wild birds (95%) were negative for all flaviviruses and no serological evidence for DENV-3, DENV-4, ZIKV, and YFV was found. In addition, 5 equines (25%) and 2 birds (2%) were classified as “unspecific”. No seroconversion events were detected. Wild birds with SLEV-neutralizing antibodies were *Empidonax virescens*, a migratory species that migrates from Canada and *Myiozetetes similis*, a resident species of Costa Rica [32].

### 3.2. There Was No Evidence of Active Infection in Birds and Mosquitoes

To study the epizootic cycle of these arboviruses, mosquitoes and wild birds were sampled. A total of 140 wild birds were collected during the period of the study. The complete postmortem and histopathological analyses showed no lesions associated with arbovirus infections. In the Cuajiniquil area, 52 wild birds from 15 different species were captured, 2 species were migratory (Appendix A). In Talamanca, 88 wild birds were captured from 29 different species, 6 species were migratory. This area is a very important point of migration from North America to South America [46].

Additionally, 1373 mosquitoes were captured in 128 trap-nights. The most frequent species sampled during this study in the Cuajiniquil area (*n* = 554) were *Deinocerites pseudes* (24.9%, *n* = 138), *Cx. quinquefasciatus* (17.7% *n* = 98), and *Anopheles albimanus* (8.7% *n* = 48). In the area of Talamanca (*n* = 819), the most frequent species sampled were *Cx. quinquefasciatus* (45.9%, *n* = 376), *Cx. coronator* (11.6%, *n* = 95), and *Mansonia titillans* (10.0%, *n* = 82). The complete classification of the mosquitoes according to the location and the sampled season is available in Appendix A. Mosquitoes were grouped according to the collection site and species (maximum 20 individuals per pool). Blood-engorged females were analyzed individually to identify their blood meal. Mosquito pools (*n* = 164 for Cuajiniquil and *n* = 198 for Talamanca), blood-engorged females (*n* = 32), and wild birds (*n* = 140) were analyzed by a seminested RT-PCR using Flavivirus genus specific primers [43]. Two mosquito pools from Talamanca were identified as positive for flaviviruses (Table 3). Positive pools were submitted for nucleotide sequencing and showed homology to a mosquito flavivirus (GenBank accession number MN856866.1 and MK241496.1). In the case of wild bird samples, no positive PCR results were obtained.

### 3.3. The Analyses of Mosquito Blood Meals Show a Species Diversity of Feeding Sources

Mosquito blood meals were analyzed to establish the diversity of blood meal sources and possible participation of these species in the putative virus cycle. Mosquitoes that were classified as blood-engorged females were taxonomically identified using morphological characters and analyzed for blood meal identification by detection of COI [45]. A total of 23 of the 32 mosquitoes led to positive DNA amplification: from Cuajiniquil: *Cx. restrictor* (*n* = 1), *Cx. quinquefasciatus* (*n* = 1), *Anopheles albimanus* (*n* = 1), and *Deinocerites pseudes* (*n* = 1) and from Talamanca: *Cx. quinquefasciatus* (*n* = 9), *Cx. coronator* (*n* = 3), *Cx. (Melanoconion)* sp. (*n* = 2)*, Psorophora ferox* (*n* = 1), *Cx. pseudostigmatosoma* (*n* = 2), and *Mansonia titillans* (*n* = 1) (Table 4).

After sequencing and BLAST analyses, we detected dog (*Canis lupus familiaris*) (*n* = 6, 31.6%), human (*Homo sapiens sapiens*) (*n* = 5, 26.3%), equine (*Equus ferus caballus*) (*n* = 4, 1.1%), sheep (*Ovis orientalis aries* ) (*n* = 4, 21.1%), and wild bird (*Columbina passerina*) (*n* = 1, 5.2%) blood used as feeding source. One sample showed a mixed blood pattern (dog/human) (Table 4).

## 4. Discussion

In this study, we detected the circulation of WNV and SLEV in two regions of Costa Rica (Cuajiniquil and Talamanca). Although we did not detect WNV or SLEV RNA in wild bird organs or mosquito pools, active circulation of those viruses was evidenced by seroconversion events at both sampling sites and by the detection of neutralizing antibodies in wild bird samples. Interestingly, our results show simultaneous circulation of several flaviviruses in the sampled areas: WNV, SLEV, and DENV-1 in Cuajiniquil and DENV-1 and DENV-2 in Talamanca.

In these areas, equines positive against WNV by IgM ELISA detection have previously been reported [28,47]. In 2009, the first clinical case of WNV was reported in a horse also from Guanacaste, and since then, new equine cases are reported annually [28,48]. Likewise, human DENV and ZIKV cases are reported annually in those areas by the National Health authorities [30]. During the years sampled for this study (2017–2018), both sites reported cases of DENV and ZIKV, most of them diagnosed by symptoms in the clinic and passive surveillance but not as laboratory-confirmed cases, situation normal for endemic countries in which it is not possible to confirm with RT-PCR every single “dengue-like” illness [30]. Interestingly, Brazil and Argentina have reported sporadic cases of SLEV in people presenting mild febrile, “dengue-like” illness, thus resulting in a misdiagnosis of the causative agent [49,50,51]. It is, therefore, tempting to speculate that human WNV and SLEV infections are being mistaken for DENV and ZIKV in endemic areas, e.g., Costa Rica.

Serological analysis showed that neutralizing antibodies against WNV and SLEV are uniformly distributed in Cuajiniquil. There, in each household, at least one of the sampled species had neutralizing antibodies (wild birds, chickens, equines, and/or humans). On the other hand, in the Caribbean sampled area, no evidence of previous contact with WNV was recorded, but serological evidence against SLEV was documented in wild birds, horses, and humans. Likewise, each household had at least one species test positive. The strikingly high seropositivity to WNV in Cuajiniquil and to SLEV in both regions reveals that these viruses might be widely distributed within Costa Rica.

We detected four wild birds with neutralizing antibodies (three for SLEV and one for WNV) belonging to four different species. Three were resident wild birds (*Campylorhynchus rufinucha*, *Myiozetetes similis*, and *Turdus grayi*) suggesting local contact with the virus and raising the possibility that the mosquito–bird–mosquito virus cycle is well established locally. The fourth one, *Empidonax virescens*, is a migratory species that was captured in Talamanca. This area is one of the most important sites for wild bird migration in the world [46]. During the yearly migration period from October to November, thousands of wild birds fly over Talamanca on their way to South America [46,52]. This migratory behavior could lead to the introduction or local emergence of wild bird-hosted flaviviruses including new strains of SLEV and WNV [19,53]. In the Americas, the role of migratory birds in the spread of WNV is not clear; however, recent studies in other continents have shown the important role of migratory birds in the introduction of new variants of WNV to the territories to which they migrate each year [54,55]. Therefore, this same role could be present in the American continent. Nevertheless, Costa Rica has never reported massive avian deaths, suggesting that the local species might be less susceptible to WNV disease.

WNV and SLEV share common mosquito vectors (*Culex*) and present comparable transmission cycles and clinical signs [15,17,56]. Costa Rica lacks information regarding which mosquitoes can be the possible vectors of WNV and SLEV (or other non-DENV or ZIKV arboviruses). However, some potential vectors are present in the country such as *Cx. quinquefasciatus*, *Cx. thriambus*, and *Cx. nigripalpus* [36]. At both sampling areas, *Cx. quinquefasciatus* was the most abundant mosquito species collected. Other species of *Culex* such as *Cx. nigripalpus* were also identified. The latter has been proposed as a vector for WNV and SLEV in America [20,57]. The blood meals in our study exemplify that *Cx. quinquefasciatus*, *Cx. Coronator*, and *Cx. pseudostigmatosoma* use humans and animals as a food source. This behavior favors virus transmission between different species, so their vectorial competence for WNV in Costa Rica should be investigated. Blood meals and species distribution further support the hypothesis that Culex species may be serving as bridging vectors capable of transmitting WNV between wild birds and final hosts, e.g., humans and equines.

In Central America, there have been no reports of outbreaks caused by SLEV or WNV in humans. However, prior studies also found serological evidence of virus circulation [58]. In vitro studies suggest that prior infection with ZIKV or DENV modulates subsequent infection with a different flavivirus and might confer cross protection [59] and Central America is hyperendemic for these two viruses. Alternatively, SLEV (or other arboviruses) could out-compete with WNV for vector and virus amplifiers [60]. The lack of outbreak reports might also be explained by less virulent strains circulating in Central America, since migrating birds carrying outbreak-causing virus do succumb during long-distance migration [53,61].

Costa Rica, like the rest of Central America, lacks information about the seroepidemiology of WNV and SLEV. In a literature review conducted by Ortiz et al., 2022, it was found that there is no recent literature or official information about the sanitary status of SLEV and WNV in Central America [58]. Our study demonstrates an ongoing circulation of WNV in the region of Cuajiniquil and SLEV in Talamanca. Our results also indicate the cocirculation of other flaviviruses such as DENV and ZIKV, and suggest that other flaviviruses could also be circulating [22]. Regions with multiple flaviviruses encounter a significant challenge in the clinical and serological diagnosis. Laboratory testing is crucial for accurate diagnosis because symptoms can overlap. Many ELISA kits are not completely devoid of cross-reactions (a necessity for accurate interpretation of results) so there is potential for misinterpretation [25]. Molecular diagnosis by RT-PCR of serum, plasma, and cerebrospinal fluid is of limited value for routine diagnosis, due to the low level and short-lived viremia generated by these viruses [25]. The PRNT ≥90% technique is the gold standard for identifying antibodies against different flaviviruses, but this technique is expensive, needs laboratory facilities, and requires careful interpretation. Thus, flavivirus serological diagnosis is indeed a real challenge [25].

Our study has some limitations. Specifically, the sample size, the limited number of areas that were sampled, and the lack of sentinel animal usage. Further studies must be focused on establishing nation-wide seroprevalence, identifying vectors and reservoirs, and identifying genotypes that might be circulating in the country. Costa Rica, as a tropical country, is susceptible to introduction and establishment of emerging and re-emerging flaviviruses that could result in an even more complex epidemiologic scenario.

Active surveillance for WNV and SLEV must be performed in flavivirus-endemic areas using mosquitoes, wild birds, and sentinel chickens to detect the viruses before re-emergence, the outburst of disease, outbreaks or even establishment of the virus in regions where all the components of the transmission cycle are present. Additionally, WNV and SLEV must be considered as a differential diagnosis in patients suspected for DENV and ZIKV infection. Here, we show that they are indeed circulating in these hyperendemic regions. Therefore, they should be considered by the health and epidemiology authorities for future prevention campaigns and arboviruses differential diagnostics.

## Figures and Tables

**Figure 1 pathogens-12-00007-f001:**
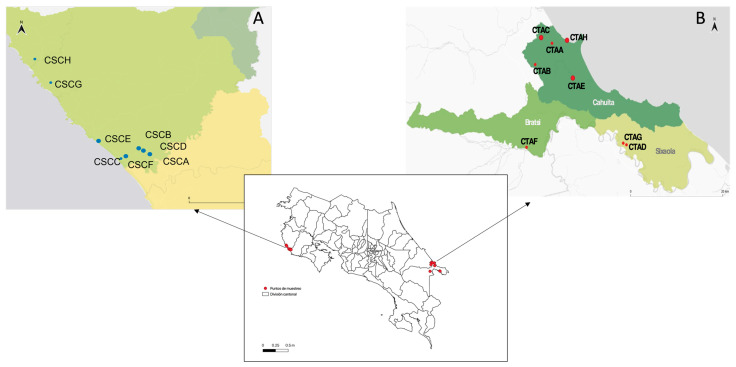
Geographic location of the sampling sites. (**A**) Cuajiniquil, geographic distribution of households, each dot represents a household. (**B**) Talamanca, geographic distribution of households, each dot represents a household. Map created using QGIS. QGIS Geographic Information System. Open-Source Geospatial Foundation Project. http://qgis.osgeo.org, accessed on 23 November 2021.

**Figure 2 pathogens-12-00007-f002:**
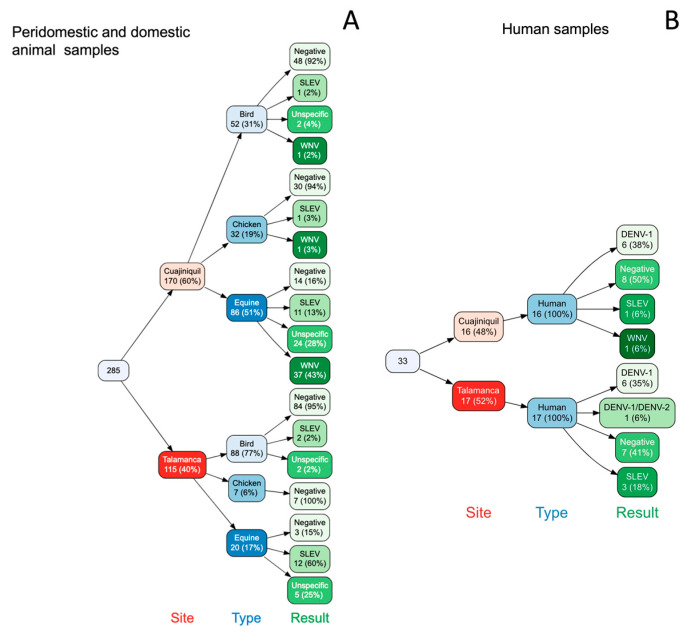
Distribution of the PRNT 90 results. (**A**) Distribution of the PRNT90 results of the sampled animals, grouped by sampling site, type, and result. (**B**) Distribution of the PRNT90 results of human samples grouped by site and result.

**Table 1 pathogens-12-00007-t001:** Type and number of samples collected at the rainy and dry season in Cuajiniquil.

Species	Sample Type	Rainy Season (*n*)	Dry Season (*n*)
Equine	Serum	86	57
Wild birds	Tissue/serum	16	36
Humans	Serum	16	12
Mosquitoes	Pools	119 (377 individuals)	44 (177 individuals)
	Blood-engorged	9	3
Chickens	Serum	29	3

**Table 2 pathogens-12-00007-t002:** Type and number of samples collected at the rainy and dry season in Talamanca.

Species	Sample Type	Rainy Season (*n*)	Dry Season (*n*)
Equine	Serum	20	16
Wild birds	Tissue/serum	60	28
Humans	Serum	17	15
Mosquitoes	Pools	133 (573 individuals)	66 (246 individuals)
	Blood-engorged	17	5
Chickens	Serum	7	0

**Table 3 pathogens-12-00007-t003:** Mosquitoes positive in flavivirus RT-PCR and their homology.

Pool Identification	Household	Species	Homology	GenBank
TCC28	CTAC	*Culex* (*Melanoconion*) sp.	Mosquito flavivirus	MN856866.1
TDC19	CTAD	*Aedes aegypti*	Aedes flavivirus	MK241496.1

**Table 4 pathogens-12-00007-t004:** Mosquito blood meals and their homology.

Pool Identification	Household	Species	Homology	GenBank
OEC7	CSCE	*Culex restrictor*	*Equus ferus caballus*	MH605334.1
OFC4	CSCF	*Anopheles albimanus*	*Homo sapiens sapiens*	MF588853.1
OFC22	CSCF	*Culex quinquefasciatus*	*Equus ferus caballus*	MG761997
OMC4	CSCM	*Deinocerites pseudes*	*Canis lupus familiaris*	KU290927
TBC1	CTAB	*Culex quinquefasciatus*	*Canis lupus familiaris*	MH105046.1
TCC1	CTAC	*Culex coronator*	*Canis lupus familiaris*	KM061528.1
TCC2	CTAC	*Culex coronator*	*Homo sapiens sapiens*	K792836.1
TCC23	CTAC	*Culex* (*Melanoconion*) sp.	*Equus ferus caballus*	MG761996.1
TCC24	CTAC	*Culex coronator*	*Equus ferus caballus*	MG761996.1
TDC8	CTAD	*Psorophora ferox*	*Homo sapiens sapiens*	K792836.1
TFC1	CTAF	*Culex quinquefasciatus*	*Ovis orientalis aries*	MG489885.1
TFC2	CTAF	*Culex quinquefasciatus*	*Homo sapiens sapiens*	MK792836.1
TFC3	CTAF	*Culex quinquefasciatus*	*Homo sapiens sapiens*	MK792836.1
TFC4	CTAF	*Culex quinquefasciatus*	*Ovis orientalis aries*	MG489885.1
TFC5	CTAF	*Culex quinquefasciatus*	*Ovis orientalis aries*	MG489885.1
TGC3	CTAG	*Culex pseudostigmatosoma*	*Canis lupus familiaris*	KU290927.1
TGC4	CTAG	*Culex pseudostigmatosoma*	*Homo sapiens sapiens*	MK103007.1
THC3	CTAH	*Culex quinquefasciatus*	*Columbina paserina*	DQ433535.1
THC6	CTAH	*Culex (Melanoconion)* sp.	*Ovis orientalis aries*	MG489885.1
TMC9	CTAM	*Culex quinquefasciatus*	*Canis lupus familiaris*	KM061528.1
TMC7	CTAM	*Culex quinquefasciatus*	*Canis lupus familiaris*	KM061528.1
TMC8	CTAM	*Culex quinquefasciatus*	*Canis lupus familiaris/ Homo sapiens sapiens*	KU696393.2/MK103007.1
TNC9	CTAN	*Mansonia titillans*	*Equus ferus caballus*	MG761996.1

## Data Availability

Data generated or analyzed during this study are included in the published article.

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
