# Peer review of "Serosurvey in Two Dengue Hyperendemic Areas of Costa Rica Evidence Active Circulation of WNV and SLEV in Peri-Domestic and Domestic Animals and in Humans"

_pathogens, 2022, doi:10.3390/pathogens12010007_

Round 1

Reviewer 1 Report

Marta Piche-Ovares and colleagues investigated the circulation of WNV and SLEV in two DENV and ZIKA hyperendemic regions in Costa Rica. Peri-domestic (wild birds) and domestic animals (chicken and equine), mosquitoes, and humans were sampled to detect the virus using PRNT and RT-PCR during rainy and dry seasons. No live viruses were detected, but several seroconversion events were found to support the active circulation of WNV and SLEV in Costa Rica.

Overall, the study is well designed, and the experimental approaches and the interpretation of the data are appropriate. But there are several points to be strengthened to make such conclusions. Some major comments/questions are found below:

1: In Introduction, please add some literature about the Flavivirus infection data in Costa Rica, especially about SLEV and WNV.

2: For the whole data presentation, the authors did not use any statistical methods to process the data.

3: In Figure 2, please add the 0 case group and lay out the data in the same order.

4: How does the author define “unspecific”?

5: How does the author explain the seroconversion cases in Cuajiniquil but not in Talamanca? What do you think makes the difference?

6: Please use the table to display the result 3.3, like mosquito species and positive amplification. 

7: Please explain what’s the meaning of the results of blood feeding source in the Discussion.

8: Some minor comments:

Line 23: extent to extents

Line 28: equine to equines

Line 45: involves to involve

Line 177: manufacturer to manufacturer’s

Line 249: flavivirus to Flavivirus

Reviewer 2 Report

It is well designed and written study. The study results are significant and provide insight on the circulation of WNV and SLEV in peri-domestic and domestic animals and humans in Costa Rica. I have a few minor suggestions

1. Heading can be improved i.e ln 255-256. Heading should be short

2. status of these viruses in neighboring countries and role of migratory birds may be emphasized as well. 
